# A Study on the Sustainable Relationship among the Green Finance, Environment Regulation and Green-Total-Factor Productivity in China

Yanhong Liu [1] , Jia Lei [2] and Yihua Zhang [3],*

1   Normal College, Shenzhen University, Shenzhen 518060, China; lucyliu@szu.edu.cn
2   Department of Urban Management, School of Government and Management, Central University of Finance and Economic, Beijing 100081, China; 2018311698@email.cufe.edu.cn
3   Department of Investment, School of Finance and Economics, Jimei University, Xiamen 361021, China
*   Correspondence: yhzhang@jmu.edu.cn; Tel.: +86-138-0601-3854

**Abstract:** Exploring the mechanism and constraints of Green Finance on high-quality economic development is of great significance to achieve the strategic goal of carbon peak and carbon neutral. Based on the panel data of 30 provinces in China from 2009 to 2019, this paper uses the epsilon-based measure model and entropy method to measure the total factor rate of green economy and the development level of green finance. It then brings green finance, technological innovation, industrial structure upgrading, environmental supervision and high-quality economic development into a unified research framework for the first time. By constructing a panel two-way fixed effect model, regulatory intermediary effect model and threshold effect model, this paper empirically tests the action mechanism and constraints between green finance and high-quality economic development. The results show that: (1) The spatial evolution of green finance in China presents a gradient decreasing pattern from east to middle to west, coastal to inland, and the spatial evolution presents an obvious southwest-northeast pattern. (2) Green finance does have a significant role in promoting high-quality economic development, in which technological innovation and industrial structure upgrading play a part of the intermediary role. This conclusion is still valid under the robustness test of lagged explanatory variables and after the possible endogenous problems are alleviated by the difference-in-difference model (DID). (3) Environmental regulation plays a non-linear regulatory role in the relationship between green finance and high-quality economic development, and there is a single threshold value. Too high intensity of environmental regulation will weaken green finance, resulting in the innovation compensation effect being more diminutive than the circular cost effect. At this time, the high-quality economic development presents a state of diminishing marginal benefits.

**Keywords:** green finance; environment regulation; green total factor productivity; moderating mediating effect; threshold effect; DID method

## 1. Introduction

At present, environmental pollution, climate change and natural resource constraints have become global topics. From the history of world economic development, the economic growth of almost all countries is accompanied by resource depletion, environmental pollution and ecological degradation. For example, the Developed Countries as United States, Britain, Germany, and the Developing Countries as China and India have experienced a circular development cycle of pollution remediation-repollution-regovernance at the cost of the environment. Although the innovation of production technology has improved the utilization rate of resources to a certain extent, this unsustainable mode of production has posed a serious threat to the ecosystem on which human beings depend. Many environmental problems such as air pollution, loss of biodiversity, soil degradation, resource depletion and sharp increase of carbon emissions have reached an irreversible critical

point [1]. With the rise of the global wave of environmental protection in the 19th century, countries began to bring the green economy into the policy agenda and internalize the external cost of the environment by using government regulation and market mechanism, in order to change from the traditional industrialization model to the green development model of ecological civilization [2]. In this "creative destruction" process, the new green supply and green demand will derive advanced green economic growth points and bring new opportunities and challenges to the world economy [3]. Therefore, it is an essential task for all countries to change the mode of economic development and decouple economic growth from environmental pollution and resource consumption.

Over the past 40 years of reform and opening-up, China's economy has consistently maintained a trend of rapid growth, its total economic output has leaped to the second in the world, and the average annual growth rate of GDP has always been maintained at more than 8%. Meanwhile, China's economy is still in the "three-phase superposition" stage of growth shift, adjustment pain and initial policy digestion [4]. Behind the miraculous development, there are objective phenomena such as environmental pollution, ecological damage and resource waste, high investment, high pollution, high energy consumption, low efficiency and extensive industrial structure is undoubtedly a stumbling block to the high-quality and sustainable development of China's economy. The imbalance between resources, environment and economic development has triggered the urgent need for China's green economic transformation. In the context of green and low-carbon economic development, green finance has become a powerful engine for practicing the concept of "two mountains" and the strategic goal of "30.60" of "emission peak," "carbon neutralization" [5]. The so-called carbon neutrality is to try to reduce carbon dioxide emissions and finally achieve the goal of net zero carbon dioxide emissions. Simply put, that is to make carbon dioxide emissions "break even." At present, China is striving to achieve carbon neutrality by 2060. However, whether out of the consideration of economic development or scientific laws, carbon dioxide emissions are bound to continue to increase for a considerable period of time before they decline steadily. Once the emission reaches the record high, it marks the possible arrival of the inflection point from increase to decrease, which is what we call "emission peak." For the core connotation of green finance, the academic community has reached a consensus: in the process of economic development, take environmental protection as an important field and support project investment and financing, operation and risk control in the field of environmental protection and energy conservation, so as to promote sustainable development. In recent years, under the guidance of several green policies issued by the government, China's green financial market has ushered in unprecedented development opportunities and showed a rapid development trend. By the end of 2020, the scale of China's green credit has exceeded the 10 trillion mark, the stock scale is the first in the world, and the stock of green bonds is 813.2 billion yuan, ranking the second in the world. The policy dividends are continuously released, showing the characteristics of "large scale, excellent structure and high quality" as a whole. The international communication capacity of green finance has been further strengthened, which shows that China's green finance development is in the ascendant. As an institutional and policy innovation to promote energy conservation, environmental protection, and green development, green finance plays in economic green transformation is a topic worthy of discussion.

In fact, green finance extends China's supply side structural reform in the financial field under the digital background. Developing Internet Finance and green finance is not only an important direction of China's financial reform and development in the future, but also an important breakthrough in financial supply side reform. Internet finance itself has a green attribute, and green finance needs the support of digital Internet technology to be further inclusive and popular. Compared with traditional finance, the most prominent characteristics of green finance are: (1) It emphasizes the living environment interests of human society. Traditional finance often emphasizes economic interests, and its operation and whole activities take economic interests as the standard of consideration. Green finance

emphasizes the interests of the living environment of human society. It takes environmental protection and the effective utilization of resources as an important standard to measure the effectiveness of its activities, and guides all economic subjects to pay attention to the natural ecological balance through their own activities. (2) It relies more on the strong support and promotion of government policies. Environmental resources are public goods, which should be provided by the government. As an operating institution, commercial banks cannot take the initiative to consider whether the lender's production or services are ecologically efficient unless there are policies and regulations. With the allocation of financial service resources as a lever, it guides the flow of funds, helps scientific and technological innovation and industrial structure optimization, and cultivates new economic growth points, so as to promote China's economy to improve quality, increase efficiency, and achieve long-term stability [6]. In this process, what is the development level of green finance and regional economy? Is there an internal relationship between green finance and scientific and technological innovation, industrial structure and regional economic development? What is its specific mechanism? Further, is there an internal relationship between environmental regulation policy and green finance and high-quality economic development? This is of great significance to China's economic leapfrog development under the new development pattern.

As a lubricant of the new normal of China's economy, green finance is the result of the evolution of the financial market from low level to a high level, and also an essential tool to promote the upgrading of the supply chain and the transition of value chain. Generally speaking, the economic boost of the financial system is mainly through "quantitative effect" [7] and "efficiency effect" [8]. The former specifically reflects the original accumulation of material capital, while the latter emphasizes the development of total factor productivity. Since the new normal in China, the disappearance of population dividends and the decline of marginal capital income have led to the gradual transition of scale economy of traditional the financial system into old age. China's economic growth has transformed from "quantitative change" to "qualitative change" period of "quality improvement and deceleration" [9]. Therefore, it is particularly essential to clarify the coupling relationship and mechanism between green finance and economic development.

Theoretically, green finance is an essential support for China's green and high-quality development under the new development pattern of the double cycle and an essential requirement for preventing systemic financial risks and climate risks. Macroscopically, the level of green finance and economic growth maintains a long-term stable relationship. By absorbing the capital of the real economy, green finance transforms it into savings, and then expands effective investment [10]. This kind of green investment formed by mobilizing savings will guide social capital to gradually transition from "three high and one surplus" enterprises to green enterprises with low pollution, low energy consumption and low emission [11], so as to achieve Industrial agglomeration [12], Structure upgrading [13], Supply-side quality improvement [14]. In other words, green finance promotes high-quality economic development, in essence, through resource allocation to promote technological progress and industrial structure upgrading. In this process, green finance plays the role of "stabilizing structure" in quality [15] and plays the role of "promoting growth" in quantity [16]. The investment demand formed by green finance will cultivate many innovative green enterprises, and the industrial chain derived from it will give birth to new economic growth points, drive social employment and consumption, and directly affect economic growth [17].

In fact, finance not only plays the core function of resource allocation and the expansion function of macro adjustment, but also plays the derivative functions of wealth redistribution, information transmission, risk trading, leading consumption and corporate governance in the process of economic development [18]. Extending to the micro level, green finance can improve the micro efficiency of China's economic development. First, green finance can fill the financing gap of green enterprises by relying on financial instruments such as green bonds and green funds, and provide sufficient fund guarantee

for enterprises to carry out green innovation activities [19]; second, the traditional micro banking theory holds that information asymmetry will lead to moral hazard and adverse selection [20], which will lead to credit rationing and strengthen the transaction cost of enterprises' economic activities. The construction of green financial system can give full play to the advantages of government information resources and degrade the search cost of enterprises [21]. Third, green finance can stimulate the green demand of consumers and help the green development "the last kilometre" [22]. Especially in the context of China accelerating the construction of a double cycle pattern based on expanding domestic demand, green consumption upgrading is significant. On the other hand, China's high-quality development can provide a sound application ecology for green finance. Meanwhile, the green transformation and development of the economy will also generate high-quality green financial service demand, promote the green financial market to innovate product structure, optimize service level and open up new development space [23]. In conclusion, their relationship is not the opposite relationship, but the circular relationship of complementary and interdependent.

So far, empirical research on green finance and economic growth mainly focuses on the direct effect, involving nation, urban agglomeration, province and county, focusing on the growth effect of Green Finance on total factor productivity [24], Green R &D investment [25], green technology innovation [26], industrial structure upgrading [27] and economic efficiency [28]. There are also studies on the spatial spillover effect [29], spatial correlation characteristics [30] and spatial structure of a financial network of green finance [31] from the perspective of space. The results show that green finance has a particular threshold effect on economic growth [32]. As long as green investment and green credit develop to a specific scale, green finance will have a growth effect and this growth effect is relatively tiny. Some scholars hold the opposite view. Based on the VECM model, some studies point out that the development scale of green finance and the allocation efficiency of green resources will have a significant inhibitory effect on the development of macro-economy [33].

Reviewing the existing literature, scholars at home and abroad have carried out multi-level discussion and empirical research on the relationship between green finance and economic development, and achieved fruitful academic results. The research can be further deepened from the following perspectives: (1) Previous studies focused on the direct effect of Green Finance on economic development, ignoring the "potential endogeneity" between macro variables, and mostly focused on green finance and industrial structure upgrading, green finance and technological innovation. There is little to comb the relationship between the four, lack of in-depth discussion on the mechanism level. However, economic development is dynamic, and the mechanism of Green Finance on economic development is not completely direct, and there are complex intermediary and regulatory roles. (2) In measuring the level of regional economic development, many studies only measure the regional GDP or fiscal revenue, and pay one-sided attention to economic growth. With the rise of TFP, it has become the most direct and effective tool to measure overall economic development. However, existing studies still ignore the factors of resources and environment in the calculation process, which can not reflect the actual level of China's high-quality economic development. (3) It is worth emphasizing that uncertainty in the direction between green finance and economic development. Previous studies only studied the threshold effect of green finance itself. They did not consider the internal mechanism of other factors on the development of the green economy, which is bound to affect the robustness and accuracy of the results. (4) Economic operation and development often have inertia and path dependence, so there is a complex dual causality between green finance and economic growth. The existing literature has not been able to deal with the endogenous problem between green finance and economic growth, which also provides an opportunity for this paper to make marginal contributions. In this paper, the difference-in-difference model (DID) is used to alleviate endogenous problems as much as possible, making the research results more credible.

To sum up, this paper attempts to analyze the internal mechanism of green finance to the high-quality economic development. In addition, to alleviate the endogenous problem as much as possible, this paper further constructs a DID model, uses exogenous policy shocks to discuss the endogenous of green finance, and more robustly evaluates the economic growth effect of green finance.

## 2. Theoretical Analysis and Research Hypothesis

### 2.1. The Direct Effect of Green Finance and High-Quality Economic Development

High-quality economic development refers to the growth model of optimizing the economic structure and transforming economic kinetic energy based on building a modern economic system, taking supply-side structural reform as the mainline, innovation, coordination, green, opening and sharing as the development concept, and taking the construction of modern economic system as the primary process, and ensuring the stability of economic growth [34]. As the "acceleration" of building a green low-carbon circular development economic system, the impact of Green Finance on high-quality economic development is mainly realized through efficient economic development, stable economic development and green economic development (As shown in Figure 1).

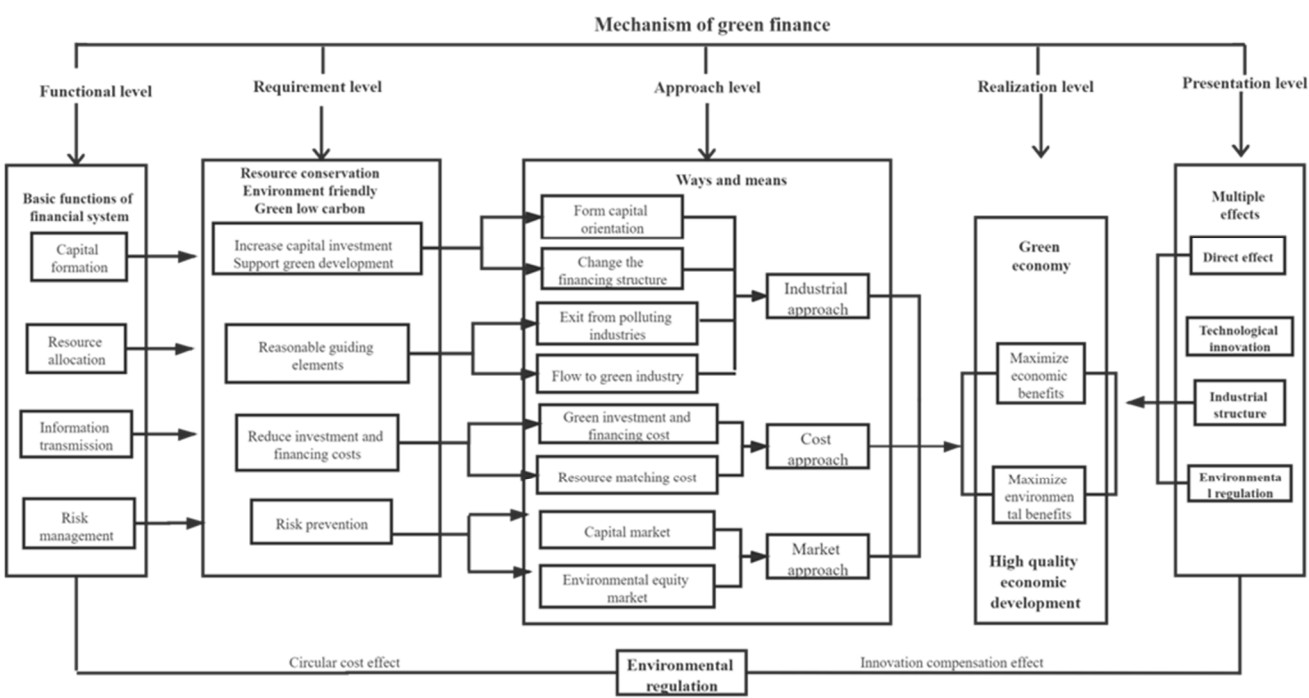

**Figure 1.** Mechanism of green finance on high-quality economic development.

From the perspective of efficient economic development, green finance can guide social capital to invest in the green industry through financial support and resource allocation effect, to reduce the capital raising cost and transaction cost in the process of green industry development, and attract more labour, social capital, land and other factor resources to further concentrate [35]; meanwhile, in the process of eliminating traditional high pollution and high energy consumption enterprises, it also stimulates the transformation and upgrading of low-end industries, effectively dissolves excess capacity [36], and improves the green total factor productivity level of enterprises. In the market-oriented economic system, green finance, through various kinds of support for the green industry, changes from the traditional mode of high-intensity resource input to the mode of high-tech and low-carbon cycle [37], promotes the development of new technology, new industry and a new model to develop new green growth level and potential supplier, and effectively improves the quality of economic development [38].

From the perspective of stable economic development, China's economic recovery and financial risks coexist after the epidemic, while investors' investment and financing decisions are affected by environmental risks [39]. Compared with traditional industries, the green industry is still immature, and its broad development prospect is bound to be accompanied by higher investment risk, which objectively increases its investment access threshold [40]. Green finance can give full play to the inherent advantages of risk redistribution, and alleviate the problems of information asymmetry and market failure by maximizing priority profit, internalizing the externality, and opportunities the risk to promote enterprises to flexibly change their business strategies [41] and improve the stability and controllability of China's economic development.

From the perspective of green economic development, the original intention of green finance is to alleviate environmental pollution, reduce waste of resources, and fight the battle of pollution prevention and control. On the one hand, green finance can inhibit pollution investment. Commercial banks regulate the capital scale of enterprises in various industries through green financial tools, reduce the amount of stock loans in high pollution industries, limit the approval and issuance of new loans in high pollution industries, limit the total capital scale, increase the loan interest rate, and increase the financial cost of pollution projects [42] to achieve the purpose of "prevention." On the other hand, green finance can encourage the development and growth of environment-friendly and low-carbon enterprises, such as green loans, financial subsidies, equity financing and other incentives to encourage new green projects, so that their positive external effects are totally endogenous to highly match the demand for pollution control [43] and achieve the purpose of "governance." Based on this, the following hypotheses are put forward:

**Hypothesis 1.** *Green finance has a direct promoting effect on high-quality economic development.*

### 2.2. The Intermediary Effect of Industrial Structure Upgrading

Under the background of carbon peak and carbon neutral, the upgrading of industrial structure is closely related to green finance. Green finance mainly realizes the upgrading and greening of industrial structure through policy orientation, capital formation, capital orientation, industrial integration and other mechanisms (As shown in Figure 2). From the perspective of policy guidance mechanism, with the preliminary establishment of a top-level design and development system of green finance, China's green finance development has rules to follow and evidence to follow, and top-down innovation and exploration are in full swing [44]. Combing the relevant policy texts, it is found that green finance mainly guides the upgrading of industrial structure through policy tools such as command, incentive and capacity-building. With the continuous tightening of environmental protection policies, the continuous improvement of support system, and the orderly expansion of pilot scope, it is expected that the policy of domestic environment supporting green finance will be increased, and the policy dividend of green transformation and upgrading of industrial structure will be released.

From the perspective of capital formation mechanism, the basic function of the financial market is to raise funds and invest in target industries, so as to form stable capital for industrial development [45]. Green finance can absorb market savings by issuing green bonds, green funds, green credit and other financial instruments, and then provide stable capital sources for the development of the green industry, form the green financial capital necessary for industrial structure transformation and upgrading, reduce the development cost of enterprises, and quickly promote the green industry to play the effect of scale agglomeration and promote the optimization and upgrading of industrial structure.

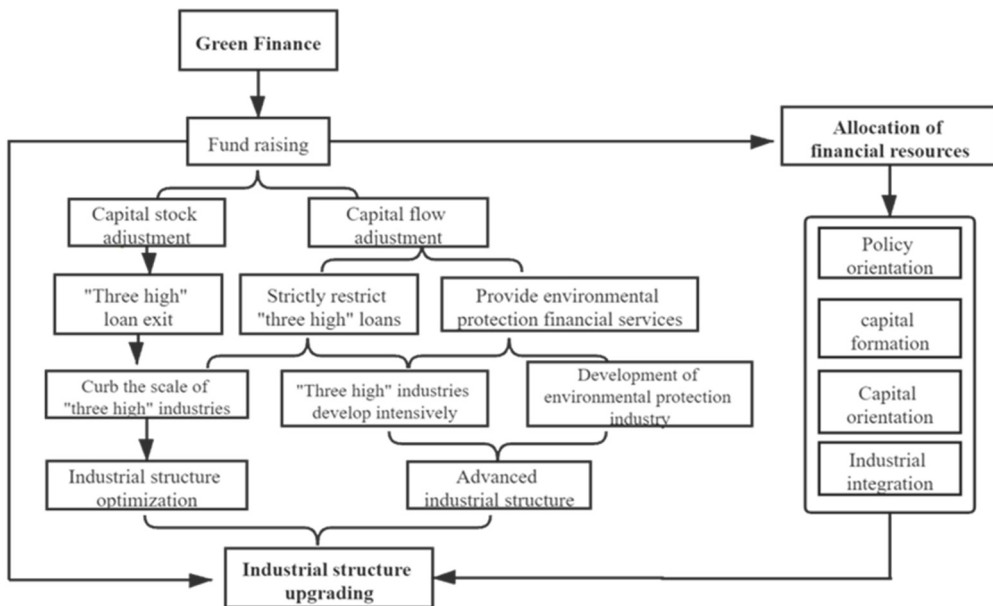

**Figure 2.** The role path of industrial structure upgrading on green finance promoting high-quality economic development.

From the perspective of capital oriented mechanism, green finance can give full play to the leverage of resource allocation and promote the free flow of financial capital among different industries, departments and regions [46]. Green finance hinders the scale expansion of "three high" enterprises by limiting loans and raising profits. It also exerts the crowding-out effect and multiplier effect of financial capital, and forces social capital to shift from industries with high pollution, high energy consumption and high emission to environment-friendly enterprises. It is bound to promote the high-end and high-quality of China's industrial structure.

From the perspective of industrial integration mechanism, green finance can give full play to the advantages of financial agglomeration. Through the industrial integration mechanism of cross-industry, cross-region and cross-ownership structure, it can reallocate resources in the process of industrial transformation, form economies of scale effect in the broader range of labour market, technology market and consumer market, and improve the competitiveness and sustainable development level of green enterprises. In this process, in addition to the flow and integration of tangible assets, intangible assets such as brand, talent, technology and the patent will also play a substantial synergistic effect, and work together to optimize and upgrade the regional industrial structure [47]. There is no doubt that the upgrading of industrial structure will directly affect the high-quality development of China's economy and form a new mechanism of economic growth. Based on it, this paper puts forward the following hypotheses:

**Hypothesis 2.** *Green finance can promote the high-quality development of China's economy by optimizing the industrial structure.*

*2.3. The Intermediary Effect of Technological Innovation*

Technological innovation is an essential driving force of high-quality economic development. Green finance mainly improves the innovation ability of enterprises through information transmission and innovation investment, and forms a new quality-oriented mechanism (As shown in Figure 3). On the one hand, the development of green technology in China is relatively late. The construction of new infrastructures such as blockchain, 5G and big data is still in its infancy, and the R&D of related technologies is facing tremendous obstacles and thresholds. Green finance tends to low polluting enterprises and green innovation projects in allocating financial funds and social resources. It supports emerging

enterprises to carry out innovation activities with the concept of green development [48]. In the process of enterprise development, the green financial system can provide sufficient financial support and financing channels, which can meet not only the capital investment required by the traditional technological innovation of enterprises, but also reserve surplus funds for the development of green technological innovation [49] to improve the technological innovation ability of enterprises, promote China's green technological innovation, and improve the level of total factor productivity. In addition, technological innovation deepens the knowledge accumulation of enterprises, resulting in the technology spillover effect, which increases the regional economic benefits.

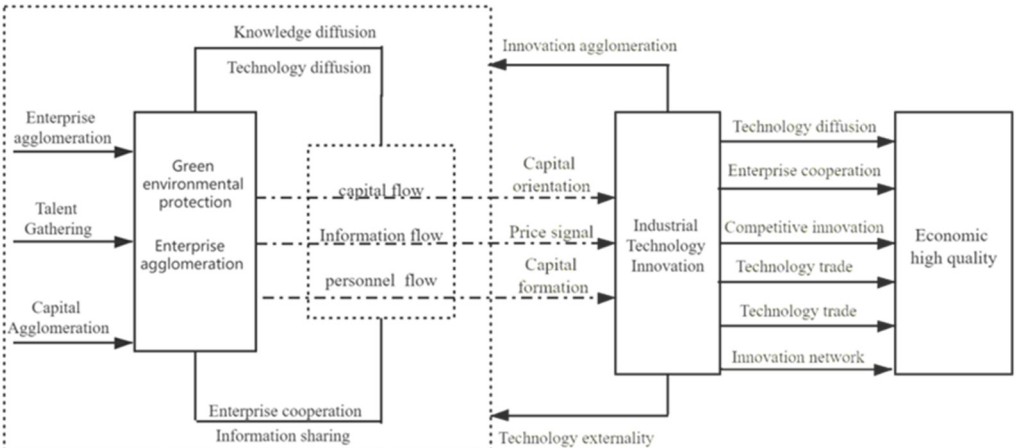

**Figure 3.** The role of technological innovation in green finance to promote high-quality economic development.

On the other hand, the uncertainty of new technology R & D and information asymmetry aggravate investors' risk aversion [50], resulting in the increase of the scale of traditional industries instead of decreasing. Green finance can give full play to the application advantages of blockchain, big data, and other technologies, reduce transaction and resource matching costs, and enhance financial institutions' risk identification and risk management capabilities. In the process of enabling financial technology, a large number of green industries will be incubated, new economic growth poles will be cultivated, and high-quality sustainable development of China's economy will be promoted. Based on this, the following assumptions are put forward:

**Hypothesis 3.** *Green finance can promote the high-quality development of China's economy through technological innovation.*

### 2.4. Regulatory Role of Environmental Regulation

In order to maintain the rapid growth of fiscal revenue, local governments take the initiative to reduce environmental protection standards, and even produce the motivation of "competition for environmental protection" for the purpose of profit [51], that is, the so-called environmental pollution effect of political resources [52]. However, as China's economy changes from a high-speed growth stage to high-quality development stage, the concept of green ecological environmental protection will run through all aspects of China's political economy and social culture. Therefore, it is reasonable to think that environmental regulation has an apparent regulatory role between green finance and high-quality economic development.

In light of domestic and international research, the impact effects of environmental regulation mainly include "the cycle cost effect" [53] and "the innovation compensation effect" [54] (As shown in Figure 4). The circular cost effect is proposed by neoclassical economics based on the perspective of static economic development. The core idea is that

environmental regulation adversely affects productivity and competitiveness by imposing constraints on enterprise behavior. From the perspective of green finance and economic development, the circular cost effect is mainly reflected in two aspects: one is that the high intensity of environmental regulation can increase the cost of enterprise pollution control, including waste recycling, pollutant discharge treatment, purification equipment purchase, etc. [55]. Meanwhile, the environmental regulation policy will force enterprises to strengthen the R & D cost of new technology and new equipment, reduce the efficiency of enterprises, resulting in some enterprises reducing the scale or even withdrawing from the market, reducing the marginal output of regional economic growth; second, environmental regulation can accelerate the "crowding out effect" of cost expenditure [56]. Under the condition of limited budget, the increase of cost expenditure of environmental protection governance is bound to crowd out the investment in new technology research and development, new talent training, new industry training and other fields, and inhibit the enthusiasm of enterprises to participate in environmental protection [57]. Therefore, from the perspective of the circular cost effect of environmental regulation, too high intensity of environmental regulation can increase enterprises' cost and reduce the industry's total factor productivity, which is not conducive to sustainable economic development.

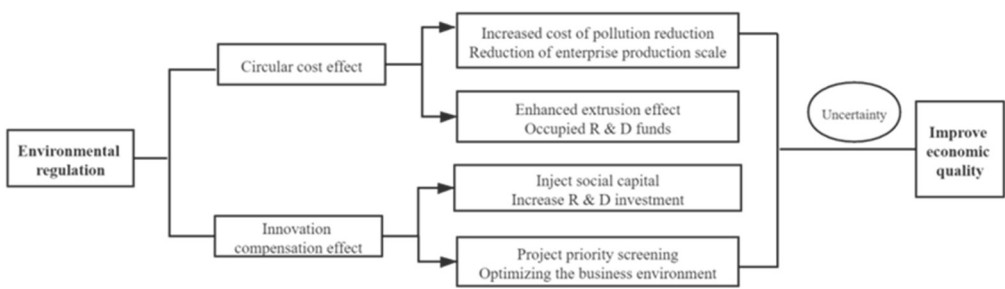

**Figure 4.** The role of environmental regulation on green finance to promote high-quality economic development.

On the contrary, Porter's innovation compensation effect thinks that in the dynamic process of economic development, environmental regulation can stimulate enterprises to innovate production mode, improve economic efficiency and offset the effect of circular cost. Specifically, with the increase of the intensity of environmental regulation, the cost loss of long-term development of enterprises is greater than the benefit output caused by technological innovation [58], which forces enterprises to increase investment in green technology research and development, improve the level of innovation, develop a green industrial chain with low pollution, low emission and high technology, and promote the upgrading of regional industrial structure [59]. In addition, with the continuous release of China's green financial policy dividend, the corresponding tax incentives and financial subsidies also emerge in an endless stream, further optimizing the business environment of the green industry [60]. From the perspective of innovation compensation effect, environmental regulation, as a win-win strategy, will force enterprises to innovate within a moderate intensity, offset the early production costs, improve production efficiency and product value [61], and ultimately drive high-quality economic development.

To sum up, appropriate environmental regulation can promote high-quality economic development through the effects of industrial exit, industrial upgrading and technological backwardness [62]. At this time, the innovation compensation effect is greater than the circular cost effect; high intensity of environmental regulation can inhibit high-quality economic development through reverse technology backpressure effect, cost effect and crowding out effect. At this time, the circular cost effect is greater than the innovation compensation effect. Based on this, the following hypotheses are put forward:

**Hypothesis 4.** *Environmental regulation has a nonlinear regulatory effect between green finance and high-quality economic development.*

## 3. Model Construction and Variable Measurement

### 3.1. Variable Selection and Measurement

### 3.1.1. Explained Variable: High-Quality Development of Regional Economy

The existing literature mainly measures the level of high-quality economic development through the index method and efficiency method. The research proves that TFP is the most direct evidence for accounting for economic growth performance [63]. It is worth noting that environmental factors have been ignored in the past studies when using TFP to measure economic quality. Under the dual carbon target, it is necessary to add energy input, pollution output and other indicators to comprehensively reflect the high-quality development degree of China's regional economy. This paper constructs the index system as shown in Table 1 to calculate each region's green total factor productivity.

**Table 1.** Measurement system of regional green total factor productivity.

| Index Nature | Index Name | Index Description |
|---|---|---|
| | labour input | Employment |
| Investment | Capital investment | Investment in fixed assets |
| | Energy input | Total energy consumption |
| Expected output | Economic output | GDP |
| | Industrial wastewater discharge | Total discharge of industrial wastewater |
| Unexpected output | Industrial waste gas emission | Total SO2 emission |
| | Discharge of industrial solid waste | Production of general industrial solid waste |

In calculation, the traditional DEA-SBM model has the disadvantages of non-angle and non-radial. When considering the unexpected output, energy consumption and pollution emissions are frequently accompanied by radial relations, while labour and capital are divisible radial relations. Given this, this paper uses maxdea7.0 software and EBM Super Efficiency mixed distance function model with radial and non radial characteristics to measure the development efficiency of economic units more scientifically and reasonably. The linear programming formula of the EBM model based on constant returns to scale and undesirable unguided output is as follows [64]:

$$\rho^* = \min \frac{\theta - \varepsilon x \sum_{i=1}^{m} \frac{W_i^- - s_i^-}{X_{i0}}}{\varphi + \varepsilon_y \sum_{r=1}^{s} \frac{W_r^+ s_r^+}{Y_{r0}} + \varepsilon_z \sum_{P=1}^{q} \frac{W_p^{z-} S_p^{z-}}{Z_{P0}}}$$

$$s.t. \begin{cases} \sum_{j=1}^{n} x_{ij}\lambda_j + s_i^- - \theta x_{io} = 0, i = 1, 2, \ldots, m \\ \sum_{j=1}^{n} y_{ij}\lambda_j - s_i^+ - \varphi y_{ro} = 0, r = 1, 2, \ldots, s \\ \sum_{j=1}^{n} y_{ij}\lambda_j + s_p^{b-} - \varphi b_{po} = 0, p = 1, 2, \ldots, q \\ \lambda_j \geq 0, s_i^- \geq 0, s_i^+ \geq 0, s_p^{b-} \geq 0 \end{cases} \quad (1)$$

Among them, $\rho^*$ is the best production efficiency value of each production decision unit; $x$. $y$ and $z$ represent input, expected output and unexpected output elements respectively; $m$, $s$ and $q$ represent the quantity of input, expected output and unexpected output respectively; $\lambda$ indicates the relative importance of the reference unit; $\theta$ and $\varphi$ represent the planning parameters of the radial part; $Sr^+$ and $S_p^{z-}$ denote the relaxation of the $r$-th expected output and the $p$-th unexpected output respectively; $W_i^-$, $W_r^+$ and $W_p^{z-}$ represent the weights of the $i$-th input, the $r$-th expected output and the $p$-th unexpected output respectively.

### 3.1.2. Core Explanatory Variable: Green Finance

In August 2016, the People's Bank of China, Ministry of Finance and other seven ministries and commissions jointly issued the "Guidance on building a green financial system", which puts forward the basic framework for the development of China's green

financial system from the aspects of green credit, green investment and green insurance. Based on the research of other scholars [65], according to the principles of availability and timing, this paper constructs a comprehensive evaluation system of regional green finance development level as shown in Table 2, and adopts the entropy method for comprehensive measurement.

**Table 2.** Comprehensive evaluation system of regional green finance development level.

| First Level Indicators | Secondary Indicators | Computing Method | Index Direction |
|---|---|---|---|
| Green credit | Sewage charge/environmental tax | Sewage charge/environmental tax | + |
| Green investment | Total investment in environmental pollution control | Total investment in environmental pollution control | + |
| Green insurance | Depth of agricultural insurance | Agricultural insurance income/total agricultural output value | + |
| Green support | Fiscal expenditure on environmental protection | Fiscal expenditure on environmental protection | + |

The entropy method is an objective weighting method for comprehensive evaluation according to the dispersion degree of an index. According to the amount of adequate information provided by the index, the index weight is determined to objectively reflect the importance of the index in the index system, that is, the greater the information entropy of the index, the greater the weight; otherwise, the smaller. Assuming that n regions are taken as samples and m evaluation indexes are selected, $x_{ij}$ represents the *j*-th evaluation index of the *i*-th region ($i = 1, 2, 3 \ldots n; j = 1, 2, 3 \ldots m$), the calculation steps are as follows:

(1) The extreme value method is used to dimensionless process the original data. The forward index and reverse index adopt the following formula respectively:

$$x'_{ij} = \frac{x_{ij} - \text{min}}{\text{max} - \text{min}} \qquad x'_{ij} = \frac{\text{max} - x_{ij}}{\text{max} - \text{min}} \tag{2}$$

where, $x'_{ij}$ is the normalized data of the *j*-th index in the *i*-th region, $x_{ij}$ is the original data, min is the minimum value and Max is the maximum value.

(2) Calculate the characteristic proportion or contribution $p_{ij}$ of the *i*-th region under the *j*-th index:

$$p_{ij} = \frac{x'_{ij}}{\sum\limits_{i=1}^{n} x_{ij}} \tag{3}$$

(3) Calculate the entropy of the *j*-th index $e_j$:

$$e_j = -\frac{1}{\ln n} \sum_{i=1}^{n} p_{ij} \ln(p_{ij}), 0 \leq e_j \leq 1 \tag{4}$$

(4) Calculate the difference coefficient $g_j$:

$$g_j = 1 - e_j \tag{5}$$

(5) Determine the weight w of the evaluation index:

$$w_j = \frac{g_j}{\sum\limits_{i=1}^{m} g_j}, \; j = 1, 2, 3 \ldots \ldots m \tag{6}$$

3.1.3. Mediating Variables

(1) Technological innovation (Innovation). Research funding is the most direct and effective tool to measure the level of regional innovation. This paper uses the degree of internal expenditure of regional research and experimental development (R&D) fund-

ing to characterize the regional technological innovation ability, and takes logarithmic processing.

(2) Upgrading of industrial structure (Industry). The upgrading of industrial structure refers to transferring the primary industry to the secondary and tertiary industries in a country or region to realize the high-quality and quantification of the economic form. This paper uses the tertiary industry and the secondary industry to measure the upgrading of industrial structure.

### 3.1.4. Regulatory Variables: Environmental Regulation (Eregulation)

Environmental regulation refers to the supervision and control of various pollution behaviors to protect the ecological environment. At present, there is no unified opinion on the measurement of environmental regulation in China, such as the number of environmental policies and regulations, the treatment rate of three wastes, etc. Considering the availability of data, this paper uses the total amount of environmental pollution treatment investment of local governments to measure, and takes logarithmic processing.

### 3.1.5. Control Variable

Foreign direct investment (Fdi) [66]. Foreign direct investment can bring advanced technology, advanced management experience, equipment, capital and talents to the region, and promote regional economic development. It is measured by FDI in different regions and treated by logarithm.

Human capital (Human) [67]. Talent reserve is essential competitiveness of regional economic development. High-quality talents can give full play to the spillover effect of knowledge, promote technology research and development, industrial upgrading, and boost economic development. The educational level of local employees measures it.

Urbanization level (Urban) [68]. Under the background of new urbanization, the urbanization of rural population can promote the transfer of primary industry to secondary and tertiary industry, promote regional coordinated development, improve social resource allocation, solve the main social contradictions in the new era, and promote high-quality economic development. It is measured by the proportion of urban population in the total population.

Economic Openness (Open) [69]. The degree of regional economic openness reflects the ability to use international market resources, give full play to comparative advantages, eliminate backward production capacity, and improve the competitiveness of local enterprises. It is measured by the proportion of total exports to GDP.

### 3.2. Construction of Regression Model

### 3.2.1. Panel Regression Model

According to the previous theory, in order to consider the direct effect of Green Finance on high-quality economic development, a two-way fixed effect model is constructed:

$$GFTP_{i,t} = \alpha_0 + \alpha_1 Gfinance_{i,t} + \alpha_2 Fdi_{i,t} + \alpha_3 Human_{i,t} + \alpha_4 Urban_{i,t} + \alpha_5 Open_{i,t} + \mu_i + \eta_t + \varepsilon_{i,t} \tag{7}$$

Among them, $i$ and $t$ represent $i$-th province and $t$-th period respectively, $\alpha_0$ is a constant term; *GFTP* is the explained variable, and *Gfinance* is the core explanatory variable; *Fdi*, *Human*, *Urban*, *Open* are a series of control variables; $\mu_i$ is individual fixed effect, $\eta_t$ is the time fixed effect, $\varepsilon_{it}$ is a random error term.

### 3.2.2. Mediating Effect Model

Taking innovation and industry as mediating variables, this paper constructs the following mediating effect model:

$$GFTP_{i,t} = \alpha_0 + c \cdot Gfinance_{i,t} + \lambda X_{i,t} + \mu_i + \eta_t + \varepsilon_{i,t} \tag{8}$$

$$Z_{i,t} = \beta_0 + \text{a} \cdot Gfinance_{i,t} + \lambda X_{i,t} + \mu_i + \eta_t + \varepsilon_{i,t} \tag{9}$$

$$GFTP_{i,t} = \chi_0 + \text{c}' \cdot Gfinance_{i,t} + bZ_{i,t} + \lambda X_{i,t} + \mu_i + \eta_t + \varepsilon_{i,t} \tag{10}$$

Among them, $Z_{it}$ is the intermediary variable, including technological innovation and industrial structure upgrading; $X_{it}$ is a series of control variables, including Fdi, Human, Urban, Open; c is the total effect of Green Finance on high-quality economic development; a is the effect of independent variable on intermediary variable; b is the effect of intermediary variable on dependent variable; c' is the effect of independent variable on dependent variable after adding intermediary variable. If ab is significant and c' is not, it is a complete mediator; if abc is significant and a × b and c have the same sign, they are partial mediators. Otherwise, they are masking effects. Other indicators are the same as Equation (7).

### 3.2.3. Regulatory Effect Model

In order to verify the complex regulatory role of environmental regulation, this paper takes environmental regulation as the regulatory variable. Based on the basic model, the interaction term and its square of environmental regulation and green finance are introduced to construct the following regulatory effect model:

$$GFTP_{i,t} = \delta_0 + \delta_1 \cdot Gfinance_{i,t} + \delta_2 \cdot Eregulation_{i,t} + \delta_3 \cdot Eregulation^2_{i,t} + \delta_4 \cdot Eregulation_{i,t}$$
$$\times Gfinance_{i,t} + \delta_5 \cdot Eregulation^2_{i,t} \times Gfinance_{i,t} + \lambda X_{i,t} + \mu_i + \eta_t + \varepsilon_{i,t} \tag{11}$$

where $Eregulation_{it}$ is the regulatory varible; $Eregulation^2_{it}$ is the quadratic term; $Eregulatio_{it} \times Gfinance_{it}$ is the interaction between the degree of environmental regulation and the green finance index. $X_{it}$ is a series of control variables, including Fdi, Human, Urban, Open. Other indicators are the same as Equation (7).

### 3.3. Data Sources

Because of the serious lack of data in Tibet Autonomous Region, this paper selects the data of 30 regions except for Hong Kong, Macao, Taiwan and Tibet from 2009 to 2019. The data are all from China Statistical Yearbook, China energy statistical yearbook, China Insurance Yearbook, China labour Statistical Yearbook, China Industrial statistical yearbook, statistical yearbooks of various regions, National Bureau of statistics, China Insurance statistical yearbook, China labour Statistical Yearbook, China Industrial statistical yearbook Wind database, etc. The missing data of individual years is supplemented by the interpolation method. The descriptive statistics of related variables are shown in Table 3.

**Table 3.** Definition and descriptive statistics of related variables.

| Type | Symbol | Name | Mean | Std | Min | Max | Observed |
|---|---|---|---|---|---|---|---|
| Explained variable | GFTP | High-Quality Economic development | 1.64 | 0.73 | 0.16 | 2.90 | 330 |
| Core explanatory variable | Gfinance | Green Finance | 0.17 | 0.10 | 0.06 | 0.79 | 330 |
| Mediating variables | Innovation | Technological innovation | 5.36 | 1.34 | 1.75 | 8.04 | 330 |
| | Industry | Upgrading of industrial structure | 1.24 | 0.69 | 0.53 | 5.23 | 330 |
| Regulatory variable | Eregulation | Environmental regulation | 5.29 | 0.85 | 2.51 | 7.26 | 330 |
| Control variables | Fdi | Foreign direct investment | 12.75 | 1.66 | 6.17 | 15.09 | 330 |
| | Human | Human capital | 9.85 | 1.12 | 7.05 | 13.90 | 330 |
| | Urban | Urbanization level | 56.43 | 12.75 | 29.89 | 89.60 | 330 |
| | Open | Economic Openness | 0.28 | 0.32 | 0.01 | 1.55 | 330 |

## 4. Empirical Analysis and Results

### 4.1. Green Financial Space Characteristics

As shown in Figure 5, based on the development index of green finance in various regions, ArcGIS 10.5 software is used to divide it into five gradients by using the natural discontinuity method to explore the development of green finance in China from the spatial level. On the whole, the development level of green finance in China is still in the primary stage, showing a gradient decreasing law from east to middle to west, coastal to inland

in space, with prominent spatial differentiation and even polarization in Beijing. In the past ten years, only Beijing is in the first gradient, and the development of green finance is far ahead. Most areas are in the third and fourth gradients, and there is a large room for growth. Most of the areas in the lower gradient are the northwest inland areas, where the level of economic development is relatively low, the industrial structure is relatively low, and the economic development still needs to be driven by the traditional industrial oriented enterprises. Most of the first and second gradients are eastern coastal provinces with a high degree of economic and social development. Green finance has a wide range of application scenarios and strong development momentum.

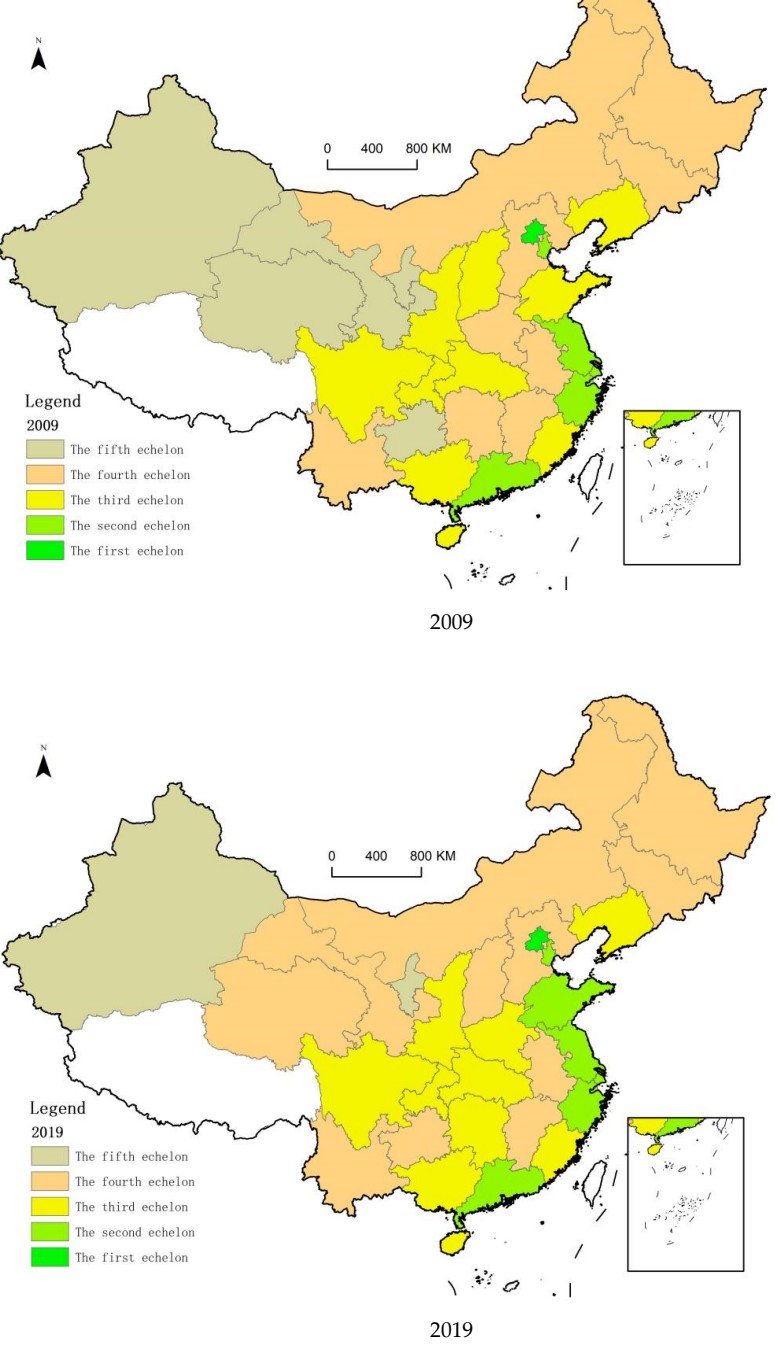

**Figure 5.** Development of green finance in 2009 and 2019.

From the perspective of spatial evolution, China's green finance has maintained a stable development trend. Gansu, Qinghai and Guizhou have leapt to the fourth gradient, Henan has leapt to the third gradient, Shandong has translated to the second gradient. In contrast, Shanxi has declined indevelopment, which is closely related to the local extensive and resource-based industrial structure. Furthermore, the spatial evolution trend of China's green finance development is analyzed with the help of the standard deviation ellipse tool, and the results are shown in Figure 6. It can be found that China's green finance has a weak evolution trend in space, showing an obvious southwest-northeast pattern of development, and has the characteristics of moving north by East. It shows that the role of green finance is deepening, and the eastern and northern regions are giving full play to the radiation effect, driving the development of green finance in inland areas.

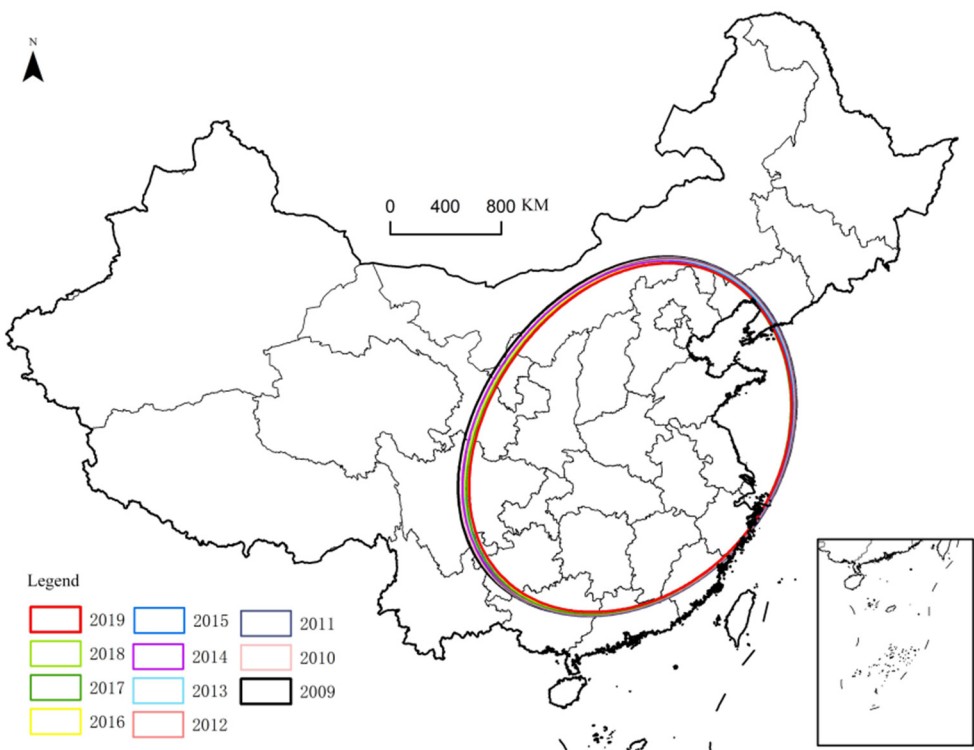

**Figure 6.** Standard deviation ellipse of China's green finance from 2009 to 2019.

*4.2. Benchmark Regression Analysis*

The panel model has mixed regression, fixed effect and random effect. Should random effect model or fixed effect model be used in regression estimation of panel data model? Through the Hausman test on the regression results of fixed effect (FE) and random effect (RE) in Table 4, it is obtained that the F statistic of Hausman test is 184.3. Rejecting the original hypothesis, which indicates that the fixed utility model (FE) is more appropriate. However, in order to ensure the robustness of the results, the results of random effect model and fixed effect model are listed at the same time, as shown in Table 4.

**Table 4.** Regression results of basic model.

| VARIABLES | Direct Effect | | | Mediating Effect | | |
|---|---|---|---|---|---|---|
| | (1)<br>FE | (2)<br>RE | (3)<br>Innovation | (4)<br>GFTP | (5)<br>Industry | (6)<br>GFTP |
| Gfinance | 0.607 *<br>(1.700) | 0.647 *<br>(1.897) | 0.818 **<br>(3.140) | 0.388 **<br>(2.594) | 1.047 ***<br>(3.299) | 0.249 ***<br>(3.528) |
| Innovation | | | | 0.268 ***<br>(13.802) | | |
| Industry | | | | | | 0.342 ***<br>(16.898) |
| Fdi | −0.004<br>(−0.216) | 0.010<br>(0.562) | 0.046 ***<br>(4.392) | −0.016 *<br>(−2.139) | −0.035 ***<br>(−6.243) | 0.008<br>(1.290) |
| Human | −0.013<br>(−0.439) | −0.013<br>(−0.434) | 0.009<br>(0.332) | −0.015<br>(−0.721) | 0.009<br>(0.266) | −0.016<br>(−1.458) |
| Urban | 0.031 ***<br>(4.357) | 0.027 ***<br>(4.152) | 0.059 ***<br>(4.728) | 0.015 ***<br>(6.927) | 0.008 ***<br>(3.415) | 0.028 ***<br>(9.086) |
| Open | −0.205*<br>(−1.907) | −0.141<br>(−1.475) | −0.570 *<br>(−2.053) | −0.052<br>(−1.313) | −0.112<br>(−0.990) | −0.167 **<br>(−2.276) |
| Constant | 7.639 ***<br>(18.295) | 7.604 ***<br>(20.312) | 1.069<br>(1.555) | 7.353 ***<br>(27.431) | 0.964 ***<br>(3.972) | 7.310 ***<br>(53.132) |
| Time effect | control | control | control | control | control | control |
| Individual effect | control | control | control | control | control | control |
| Observations | 330 | 330 | 330 | 330 | 330 | 330 |
| R-squared | 0.970 | 0.955 | 0.928 | 0.979 | 0.735 | 0.980 |
| Number of region | 30 | 30 | 30 | 30 | 30 | 30 |
| F | 184.3 *** | | 439.05 *** | 634.49 *** | 174.38 *** | 13984.38 *** |

Note: *** $p < 0.01$, ** $p < 0.05$, * $p < 0.1$.

In Table 4, columns (1) and (2) show the direct effect of Green Finance on high-quality economic development. The results show that whether using fixed effect model or random effect model, it is found that the influence coefficient of core explanatory variables is significantly positive at the 10% confidence level, indicating a significant positive correlation between green finance and high-quality economic development. The higher the degree of green finance development, the stronger its resource allocation and risk response ability to economic regulation and related industrial development to promote economic efficient, stable and green development faster and better, and assist China's high-quality economic development. Hypothesis 1 is verified. In addition, in the control variables, the impact of urbanization level on high-quality economic development is positive and significant, which indicates that in China's vigorous promotion of rural revitalization and new urbanization, the urban-rural dual structure is gradually broken, the process of agricultural population stabilization, urban-rural integration and urban agglomeration construction is accelerating, and Rural Revitalization and new urbanization are driven by two wheels, Promoting high-quality economic and social development; the impact of opening up is more significant, but it has a negative effect. In the past century, COVID-19 and Sino US trade war have exacerbated the instability of China's economy. Over-reliance on exports will have a negative impact on the healthy and stable development of China's economy. Foreign direct investment and human capital have no significant effect on high-quality economic development.

In Table 4, columns (3) and (4) show the mediating effect of technological innovation on high-quality economic development. Among them, column (3) reports the regression effect of Green Finance on technological innovation. The result shows that under the 5% confidence level, the regression coefficient of Green Finance on technological innovation is 0.818, which has a significant promoting effect. Column (4) based on column (1), the intermediate variable is added to test the direct effect of Green Finance on high-quality economic development. In the regression results, technological innovation and high-quality economic development are significantly positively correlated at 1% confidence level with a coefficient of 0.268, while green finance and high-quality economic development are significantly positively correlated at 5% confidence level with a coefficient of 0.388. Corresponding to the mediating effect model constructed above, in the path of green finance

promoting high-quality economic development, the mediating effect of technological innovation as a mediating variable is 0.219 (0.818 0.268), accounting for 56.5%, and the growth effect of green finance through technological innovation is 56.5%. To sum up, technological innovation plays a partial intermediate role in the high-quality development of green finance and economy, and hypothesis 2 is verified.

In Table 4, columns (5) and (6) show the intermediate effect of industrial structure upgrading on high-quality economic development. Among them, column (5) reports the regression effect of Green Finance on the upgrading of industrial structure. The results show that at the significant level of 1%, the regression coefficient of Green Finance on the upgrading of industrial structure is 1.407, which has a significant role in promoting. Column (6) adds the intermediary variable of industrial structure upgrading on the basis of column (1). In the regression results, industrial structure upgrading and high-quality economic development are significantly positively correlated at 1% confidence level with a coefficient of 0.342. In comparison, green finance and high-quality economic development are significantly positively correlated at 1% confidence level with a coefficient of 0.249. In green finance, industrial structure upgrading and high-quality economic development, the total effect is 0.607, the direct effect is 0.249, and the intermediary effect is 0.481 (1.407 0.342). In the regression results of the whole sample, the upgrading of the industrial structure plays a partial intermediary role in the high-quality development of green finance and economy, and hypothesis 3 is verified.

### 4.3. The Nonlinear Effect of Environmental Regulation

Firstly, environmental regulation is added to the benchmark regression model to observe the dual marginal impact of environmental regulation on high-quality economic development. Columns (1) and (2) of Table 5 report the estimated results after joining environmental regulation and its secondary items. High quality economic development is a comprehensive expression of various factors such as green development, rationalization of industrial structure, intensity of economic growth and stability of economic growth. Appropriate environmental regulation can stimulate enterprises' innovation behavior, promote the improvement of production efficiency, reduce enterprises' production costs, and have a signal effect on enterprises' inefficient resource investment and technological innovation, Forcing them to carry out technological innovation and industrial structure upgrading activities can reduce the level of environmental pollution in this process, so as to promote the Pareto improvement of the whole society, and have the forced effect of "innovation compensation" for high-quality economic development. Nevertheless, once environmental regulation exceeds the inflection point level, excessive environmental regulation will limit high-quality economic development. The high environmental externality governance cost caused by human economic production activities reduces the production efficiency and product profits of enterprises, and has a retrogressive effect of "cost compliance" on high-quality economic development. It is found that, at the 1% confidence level, the first term coefficient of environmental regulation is significantly positive, while its square term is significantly negative correlated with high-quality economic development, which reflects that environmental regulation will have an inverted U-shaped mechanism on high-quality economic development; that is, the trend of economic development first rises and then falls. It shows that the control of environmental regulation can promote high-quality economic development. On the contrary, environmental regulation should not exceed a certain range. Otherwise, there will be policy inhibition, and the inflection point of the inverted U-curve will appear in advance.

**Table 5.** Regulatory effect model results.

| VARIABLES | (1) GFTP | (2) GFTP | (3) GFTP |
|---|---|---|---|
| Gfinance | 0.571 *** | 0.860 ** | −4.500 ** |
| | (3.183) | (2.294) | (−2.859) |
| Eregulation | 0.055 *** | 0.364 *** | 0.267 * |
| | (4.327) | (3.892) | (1.822) |
| Eregulation 2 | | −0.030 *** | −0.012 |
| | | (−3.711) | (−0.871) |
| Gfinance×Eregulation | | | 2.098 *** |
| | | | (3.441) |
| Gfinance×Eregulation 2 | | | −0.194 *** |
| | | | (−3.414) |
| Urban | 0.026 *** | 0.025 *** | −0.001 |
| | (9.100) | (4.008) | (−0.197) |
| Human | −0.011 | −0.013 | −0.010 |
| | (−0.631) | (−0.492) | (−0.573) |
| Open | −0.157 ** | −0.110 | 0.030 *** |
| | (−2.379) | (−1.121) | (8.561) |
| lnfdi | −0.009 | 0.001 | −0.161 |
| | (−1.105) | (0.102) | (−1.689) |
| Constant | 7.668 *** | 6.781 *** | 7.572 *** |
| | (34.074) | (14.858) | (33.421) |
| Observations | 330 | 330 | 330 |
| R-squared | 0.972 | 0.975 | 0.9726 |
| Number of region | 30 | 30 | 30 |

Note: *** $p < 0.01$, ** $p < 0.05$, * $p < 0.1$.

Meanwhile, the interaction term between green finance and environmental regulation is added to the benchmark regression model to analyze how the marginal impact of Green Finance on high-quality economic development will change with the change of environmental regulation intensity, and further test the regulatory effect of environmental regulation on green finance and high-quality economic development. The results are shown in column 3 of Table 5. It can be found that the interaction coefficient between green finance and environmental regulation is significantly positive at the significant level of 1%, which indicates that the interaction effect between green finance and environmental regulation will promote high-quality economic development, and the innovation compensation effect of environmental regulation is greater than the circular cost effect; however, the interaction coefficient between green finance and environmental regulation is significantly negative, which indicates that environmental regulation plays a negative regulatory role between green finance and high-quality economic development. That is, the interaction effect between them will inhibit high-quality economic development. This result means that with the increase of environmental regulation intensity, the positive effect of Green Finance on high-quality economic development shows an obvious inverted U-shaped change trend of first strengthening and then weakening. Too high environmental regulation intensity will make the circular cost effect greater than the innovation compensation effect, and weaken the effect of Green Finance on high-quality economic development. Environmental regulation is weak in the early stage of environmental regulation and green finance development, and the government mostly adopts incentives, capacity-building, and other policy tools. Environmental regulation plays a strong positive regulatory role between green finance and high-quality economic development; with the increase of supervision, this positive regulatory role will continue to weaken or even distort. Thus, it is verified that environmental regulation plays a nonlinear regulatory role between green finance and high-quality economic development, and Hypothesis 4 holds.

*4.4. Endogenous Test*

Endogeneity is a common and difficult problem in macroeconomic research. The existence of endogeneity will lead to the error of model estimation results. When studying the impact of Green Finance on high-quality economic development, although we control many variables, there may still be endogenous problems caused by the omission of variables and the causal relationship between explanatory variables and explained variables, which will lead to the rationality of the model results (using Durbin Wu Hausman for endogenous test, the p value of F statistic is less than 0.05, It indicates that there is strong endogeneity). The endogeneity of this paper may be caused by the following two aspects: first, there are missing variables in the model. Green total factor productivity includes economic, resource and environmental factors. It is a comprehensive index with many influencing factors. Therefore, some variables may be omitted from the model. This paper has tried to increase the control variables, and used the two-way fixed effect model to control the unobserved individual effects and time effects in the region, so as to remove the unobservable individual heterogeneity effects as much as possible, to reduce the errors caused by the research. Second, there is a two-way causal relationship between green finance and high-quality economic development. Due to the inertia and path dependence of economic operation and development, there is not a one-way causal relationship between green finance and economic growth. The development of green finance can promote economic growth, and sustainable economic growth in turn will provide a good development environment for green finance. Generally, areas with better economic development have stronger technological innovation ability and human resources advantages, which may develop or attract more advanced green financial capital and technology. For areas with lower green financial development level, the government may give preferential policies and funds to increase the construction and investment of technical infrastructure. To sum up, it is possible that the model may have strong endogeneity due to the serious two-way causality between the two.

Based on this, this paper attempts to use the difference-in-difference method to solve the possible endogenous problems in the model, so as to make the empirical analysis results more accurate. Because regional economic growth is often affected by many factors, in order to more steadily evaluate the economic growth effect of green finance, this paper uses exogenous policy impact to discuss the endogenous nature of green finance. The basic idea of double difference is to take the samples affected by the policy as the treatment group and the samples not affected by the policy as the control group based on the counterfactual framework. The effect after the implementation of the policy is evaluated by comparing the differences between the treatment group and the control group. Further, the pilot green finance policy is exogenous, and the implementation of the pilot policy is regarded as a "quasi-natural experiment." In 2017, the executive meeting of the State Council of the people's Republic of China decided to build green finance reform and innovation pilot zones with emphasis and characteristics in five provinces (regions) such as Zhejiang, Jiangxi, Guangdong, Guizhou and Xinjiang, Explore replicable and replicable experience in system and mechanism. After the establishment of the green financial reform and innovation pilot zone, the change of regional green development performance mainly comes from two parts: one is the "time effect" formed based on time natural growth or economic development; the second is the "policy effect" caused by the approval of the green financial reform and innovation pilot zone. Since the influence intensity of policies is different between the selected areas and the non-selected areas, the selected cities can be regarded as the experimental group and the non-selected cities as the control group. The basic assumption model is as follows:

$$GFTP_{i,t} = \theta_0 + \theta_1(\text{treat}_i \times \text{time}_t) + \lambda X_{i,t} + \mu_i + \eta_t + \varepsilon_{i,t} \qquad (12)$$

where subscripts *i* and *t* represent the region and year respectively, and *GFTP*$_{it}$ is the explanatory variable, indicating the green total factor productivity of the *i*-th region in *t*-th year; treat $\times$ time is the core explanatory variable, which represents the dummy variable of

the approved green financial reform and innovation experimental area. Among them, if the region is set up as a pilot area for green financial reform and innovation, then treat is 1, otherwise, it is 0; time is the time identification variable of the pilot area, which is 0 before 2017 and 1 after 2017. Its coefficient $\theta_0$ is the core parameter to be evaluated, indicating the net effect of green finance pilot policy on Regional Green total factor productivity. Other indicators are the same as Equation (7).

An essential premise of DID model is to meet the parallel trend hypothesis to ensure the unbiased nature of the estimator. In order to test the appropriateness of DID model, with the help of event research method, focusing on the year when the green financial reform and innovation pilot zone was approved, we investigated whether there was a significant difference in the changing trend of economic quality between pilot areas and non pilot areas in the first three years and the second two years of policy implementation. For this purpose, the following measurement models are set:

$$GFTPi, t = \omega_0 + \sum_{2009}^{2019} \omega_t \text{treat}_i \times \text{time}_t + \lambda X_{i,t} + \mu_i + \eta_t + \varepsilon_{i,t} \tag{13}$$

where $\text{time}_t$ is the year dummy variable ($t$ = 2009, 2010, ... , 2019), $\omega_{2017}$ is the pilot area for green financial reform and innovation, and the effect of the implementation of the pilot strategy in that year, $\omega_{2009}$ to $\omega_{2016}$ is the effect of each year before the implementation of the policy, $\omega_{2018}$ to $\omega_{2019}$ is the effect of each year after the implementation of the policy.

Taking 2017 as the base year, the pilot year of the green financial reform and innovation pilot zone, this paper makes a separate regression on the explained variables in the first three years and the second two years and above of the base year. If the regression coefficient in the first three years is not significant and the regression coefficient in the next two years is significant, it indicates that the parallel trend hypothesis is tenable, and $\omega_{2017}$ to $\omega_{2019}$ indicates the dynamic effect on the high-quality development capacity of the regional economy with the passage of policy implementation time. The regression results (Table 6) show that whether the control variable is added or not, the coefficients of regional green total factor productivity in the first three years of the pilot of green financial reform and innovation pilot zone are not significant, and the regression coefficients are near 0, while the treat×time coefficients after 2017 are significantly positive, indicating that the experimental group and the control group have the same trend before the pilot, meeting the assumption of parallel trend. Further, from the dynamic effect of parallel trend test, the regression coefficient of regional green total factor productivity is gradually significant after the base year and generally maintains an upward trend, indicating that the pilot of green financial reform and innovation has a significant positive impact on the high-quality development of regional economy and the accelerated release of policy dividends.

**Table 6.** Dynamic regression results.

| VARIABLES | (1) GFTP | (2) GFTP |
|---|---|---|
| Before 3 | 0.042 | 0.034 |
| | (1.056) | (1.076) |
| Before 2 | 0.068 | 0.056 |
| | (1.342) | (1.501) |
| Before1 | 0.077 | 0.058 |
| | (1.307) | (1.450) |
| Current | 0.098 | 0.073 * |
| | (1.527) | (1.770) |
| After1 | 0.120 * | 0.081 * |
| | (1.754) | (1.810) |

**Table 6.** *Cont.*

| VARIABLES | (1) GFTP | (2) GFTP |
|---|---|---|
| After2 | 0.126 * | 0.077 * |
| | (1.750) | (1.751) |
| Control variable | No | control |
| Constant | 9.034 *** | 7.868 *** |
| | (493.492) | (18.754) |
| Time effect | control | control |
| Individual effect | control | control |
| Observations | 330 | 330 |
| R-squared | 0.958 | 0.970 |
| Number of region | 30 | 30 |

Note: *** $p < 0.01$, * $p < 0.1$.

After the parallelism trend test is passed, the model 12 is regressed by using the double difference method. The results are shown in Table 7. The first column is the regression results without adding control variables, and the second column is the regression results after adding control variables.Treat × time coefficients are significantly positive at the confidence level of 1%, which shows that the did regression results are highly consistent with the benchmark regression results. In conclusion, after controlling endogenous problems, green finance will indeed play a positive role in promoting the high-quality development of regional economy.

**Table 7.** DID Test results of Green Finance.

| VARIABLES | (1) GFTP | (2) GFTP |
|---|---|---|
| treat×time | 0.091 * | 0.058 * |
| | (1.829) | (1.939) |
| Human | | −0.012 |
| | | (−0.423) |
| Urban | | 0.028 *** |
| | | (4.273) |
| Open | | −0.301 *** |
| | | (−3.178) |
| lnfdi | | −0.002 |
| | | (−0.103) |
| Constant | 9.034 *** | 7.838 *** |
| | (492.728) | (19.165) |
| Time effect | control | control |
| Individual effect | control | control |
| Observations | 330 | 330 |
| Number of region | 30 | 30 |
| R-squared | 0.957 | 0.969 |

Note: *** $p < 0.01$, * $p < 0.1$.

### 4.5. Robustness Check

According to the model constructed in the previous paper, considering the possible "path dependence" effect of Green Finance on high-quality economic development, this study lagged all the explained variables for one period. The results are shown in Table 8. Except for the change of coefficient, the explanatory variables remained significant in the 10% confidence interval, which further verified the robustness of the empirical outcome.

**Table 8.** Robustness test results.

| VARIABLES | Direct Effect | | | Mediating Effect | | | Regulatory Effect | |
|---|---|---|---|---|---|---|---|---|
| | (1) FE | (2) RE | (3) Innovation | (4) GFTP | (5) Industry | (6) GFTP | (7) GFTP | (8) GFTP |
| Gfinance | 0.602 *** | 0.653 *** | 0.521 * | 0.479 *** | 1.177 *** | 0.220 *** | 0.768 *** | −2.318 ** |
| | (8.663) | (3.554) | (1.904) | (5.470) | (4.362) | (3.356) | (17.015) | (−2.638) |
| Innovation | | | | 0.284 *** | | | | |
| | | | | (14.891) | | | | |
| Industry | | | | | | 0.325 *** | | |
| | | | | | | (17.128) | | |
| Eregulation | | | | | | | 0.329 *** | 0.303 *** |
| | | | | | | | (4.358) | (3.219) |
| Eregulation 2 | | | | | | | −0.028 *** | −0.016 * |
| | | | | | | | (−4.399) | (−1.926) |
| Gfinance×Eregulation | | | | | | | | 1.295 *** |
| | | | | | | | | (3.458) |
| Gfinance×Eregulation 2 | | | | | | | | −0.124 *** |
| | | | | | | | | (−3.389) |
| Fdi | −0.005 | 0.009 | 0.030 | −0.019 *** | −0.021 ** | 0.001 | −0.000 | −0.003 |
| | (−1.475) | (1.032) | (1.426) | (−5.414) | (−3.063) | (0.422) | (−0.087) | (−0.884) |
| Human | −0.023 | −0.027 | 0.067 | −0.035 | −0.021 | −0.016 | −0.021 | −0.020 |
| | (−1.006) | (−0.889) | (1.565) | (−1.603) | (−0.630) | (−0.877) | (−1.072) | (−0.832) |
| Urban | 0.032 *** | 0.028 *** | 0.065 *** | 0.012 *** | 0.011 *** | 0.028 *** | 0.027 *** | 0.032 *** |
| | (10.289) | (4.422) | (5.377) | (3.974) | (5.257) | (9.027) | (12.244) | (9.368) |
| Open | −0.204 * | −0.135 | −0.680 *** | 0.020 | −0.113 | −0.167 ** | −0.132 | −0.182 * |
| | (−1.968) | (−1.263) | (−3.497) | (0.506) | (−0.953) | (−2.447) | (−1.497) | (−1.933) |
| Constant | 8.258 *** | 8.317 *** | 0.778 | 8.090 *** | 0.297 | 8.161 *** | 7.449 *** | 8.082 *** |
| | (23.401) | (19.278) | (0.886) | (24.035) | (1.059) | (25.147) | (14.716) | (20.875) |
| Time effect | control | control | control | control | control | control | control | control |
| Individual effect | control | control | control | control | control | control | control | control |
| Observations | 300 | 300 | 300 | 300 | 300 | 300 | 300 | 300 |
| Number of groups | 30 | 30 | 30 | 30 | 30 | 30 | 30 | 30 |
| R-squared | 0.9696 | 0.2095 | 0.9334 | 0.9793 | 0.7129 | 0.9800 | 0.9727 | 0.9712 |

Note: *** $p < 0.01$, ** $p < 0.05$, * $p < 0.1$.

## 5. Further Analysis

Based on the previous analysis, since environmental regulation will have a nonlinear regulatory effect on green finance and high-quality economic development, this paper further uses the panel threshold model to build the following model, taking environmental regulation as the threshold variable to explore its threshold effect between green finance and high-quality economic development:

$$GFTP_{i,t} = \sigma_0 + \sigma_1 Gfinance_{i,t} I(Eregulaiton \leq \varsigma_1) + \sigma_2 Gfinance_{i,t} I(\varsigma_1 \leq Eregulaiton \leq \varsigma_2) + \sigma_3 Gfinance_{i,t} I(Eregulaiton \geq \varsigma_3) + \lambda X_{i,t} + \mu_i + \eta_t + \varepsilon_{i,t} \tag{14}$$

Among them, *GFTP* is the explained variable in the formula; *Gfinance* is the explanatory variable, and *Eregulation* is the threshold variable; $\sigma_1, \sigma_2 \dots, \sigma_n$ is the green finance variable coefficient under *n* different environmental regulation thresholds, and *I* (*) is the indicator function. When the conditions in parentheses are met, it is taken as 1, otherwise it is 0; $\xi_1, \xi_2, \dots, \xi_n$, is the threshold of n different levels. Threshold regression model analysis needs to answer two questions. First, whether the threshold effect is significant; second, whether the threshold estimate is close to the real value.

Before threshold regression, we first verify whether the model has thresholds, then determine the number of thresholds, and use the bootstrap method to simulate sampling 300 times to search the threshold. The results are shown in Table 9. The threshold model of environmental regulation does not pass the double and triple threshold test, but only passes the single threshold test. Therefore, green finance has a single threshold effect on high-quality economic development with environmental regulation as the threshold variable, and the threshold value is 6.2157.

**Table 9.** Panel threshold effect test results.

| Threshold Attribute | F | P | Threshold Estimate | BS | 10% | 5% | 1% | 95% Conf. Interval |
|---|---|---|---|---|---|---|---|---|
| Single | 26.17 | 0.08 *** | 6.2157 | 300 | 23.8789 | 30.3597 | 39.2015 | [6.1363, 6.2212] |
| Double | 14.37 | 0.3133 | - | 300 | 21.1202 | 25.1659 | 35.2131 | - |
| Triple | 16.17 | 0.3567 | - | 300 | 34.2759 | 44.0404 | 61.1833 | - |

Note: *** $p < 0.01$.

It can be seen from Table 10 that in the different threshold ranges of environmental regulation, green finance plays different roles in high-quality economic development. When the intensity of environmental regulation is lower than the threshold of 6.2157, the regression coefficient of green finance to high-quality economic development is 2.08, which is significant at 1% confidence level; when the environmental intensity is higher than the threshold of 6.2157, the regression coefficient of green finance to high-quality economic development drops to 1.75, which further shows that when the environmental regulation reaches a certain intensity, it will weaken the boosting effect of Green Finance on high-quality economic development, and verify the complex regulatory effect of environmental regulation on green Finance and high-quality economic development.

**Table 10.** Panel threshold regression results.

| Variable | Coef. | Std. Err. | T | P > |t| | 95% Conf. Interval |
|---|---|---|---|---|---|
| lnfdi | 0.01 | 0.0164896 | 0.68 | 0.501 | [−0.0224793, 0.0449707] |
| Human | 0.15 *** | 0.0236942 | 6.32 | 0.000 | [0.1012534, 0.1981736] |
| Urban | 0.05 *** | 0.0046002 | 10.16 | 0.000 | [0.0373324, 0.0561494] |
| Open | −0.09 | 0.0962605 | −0.93 | 0.359 | [−0.2866172, 0.1071326] |
| Gfinance(Eregulation ≤ 6.2157) | 2.08 *** | 0.3744715 | 5.55 | 0.000 | [1.312972, 2.844732] |
| Gfinance(Eregulation > 6.2157) | 1.75 *** | 0.2922143 | 6 | 0.000 | [1.156559, 2.351849] |
| _cons | 5.05 *** | 0.2622879 | 19.26 | 0.000 | [4.516344, 5.589222] |

Note: *** $p < 0.01$.

## 6. Conclusions and Suggestions

This paper puts forward assumptions through theoretical deduction, constructs a panel two-way fixed effect model with the help of the data of various regions from 2009 to 2019, and empirically studies the mechanism of Green Finance on high-quality economic development. The study found that:

First, the development of green finance in China is generally in its infancy, and there is still great development potential. In space, it presents the gradient decreasing law of East→middle→West, coastal→inland, and there is a weak southwest northeast evolution trend. In the past 10 years, only Beijing has been in the first gradient, and the development of green finance is far ahead. Most areas are in the third and fourth gradient, and there is a large room for rise. Most of the areas in the lower gradient are northwest inland areas. Economic development level of these areas is relatively low and the industrial structure is relatively low. The economic development still needs to be driven by traditional industrial oriented enterprises. Most of the areas in the first and second gradients are coastal eastern provinces with a high degree of economic and social development. Green finance has a wide range of application scenarios and strong development momentum.

Second, green finance plays an obvious direct role in promoting high-quality economic development. The higher the development degree of green finance, the stronger its resource allocation and risk response ability to economic regulation and related industrial development, so as to promote economic, efficient, stable and green development faster and better. This conclusion is still valid under the robustness test of lagged explanatory variables and after the possible endogenous problems are alleviated by the DID model.

Third, technological innovation is an essential driving force for high-quality economic development. Green finance can provide sufficient financial support and financing channels for enterprises through information transmission and innovation investment, so as to reduce transaction and resource matching costs and enhance the risk identification and risk management ability of financial institutions. It can not only meet the capital investment required by enterprises' traditional technological innovation, but also form a new quality oriented mechanism to reserve surplus funds for green technological innovation. In the process of financial science and technology empowerment, a large number of green industries will be incubated, new economic growth poles will be cultivated, and high-quality and sustainable development of China's economy will be promoted.

Fourth, we will continue to expand the scope of sustainable financial structure and enhance the green financial system in accordance with the policies that can be explained and improved in an orderly manner, and continue to tighten the scope of green financial structure and environmental protection policies that can be supported in an orderly manner; green finance can absorb market savings by issuing green bonds, green funds, green credit and other financial instruments to form green financial capital necessary for the transformation and upgrading of industrial structure; green finance hinders the scale expansion of three high enterprises by implementing measures such as quota, loan restriction and interest withdrawal, gives play to the crowding out effect and multiplier effect of financial capital, and forces social capital to shift from industries with high pollution, high energy consumption and high emission to environment-friendly enterprises; green finance can give full play to the advantages of financial agglomeration, realize the reallocation of resources in the process of industrial transformation through the industrial integration mechanism of cross industry, cross region and cross ownership structure, and form economies of scale in a wider range of labor market, technology market and consumer market. To sum up, green finance passes the government policy oriented, capital formation, capital oriented, industrial integration and other mechanisms to realize the advanced and green industrial structure, so as to promote the sustainable development of China's economy.

Fifth, environmental regulation plays a nonlinear regulatory role between green finance and high-quality economic development. Maintaining moderate environmental regulation can make the innovation compensation effect greater than the circular cost effect, and environmental regulation plays a strong regulatory role between them; with the increase of supervision, the circular cost effect will break through the innovation cost effect. At this time, this positive regulatory effect will continue to weaken or even distort. The panel threshold effect model of environmental regulation further confirms the nonlinear regulation of environmental regulation. When the intensity of environmental regulation is controlled at a certain threshold, green finance will maintain higher marginal benefits for high-quality economic development.

Under the guidance of the new development concept, promoting green finance and high-quality economic development is the development direction of "green finance 2.0." Especially in the extreme background of increasingly severe climate change and the increasing international trade risk, China's existing green finance cannot meet the construction needs of domestic and international double cycles. Green finance still has a long way to go. Our finding on the relationship between green finance and high-quality economic development can contribute to the balance between economic development and environmental sustainability, so as to solve the increasingly serious climate change and environmental problems and strengthen the construction of a community of shared future for mankind. The policy enlightenment of this paper includes: first, continue to strengthen the policy support of green finance, cultivate the awareness of green consumption of individuals and enterprises, develop more green financial products and expand the application scenarios. Second, the theoretical model of this paper combines the research of Green finance with economic growth theory, which provides an important directional reference for promoting enterprise technological innovation, green transformation and upgrading, and improving the strategic goal of total factor productivity. We should pay attention to technological inno-

vation and industrial structure upgrading, encourage enterprises to strengthen technology research and development, reduce pollution emissions, resolve excess capacity, strengthen green investment concept and develop green projects such as carbon and environmental protection, and promote the transformation of "green water and green mountain" into "invaluable asset". Third, in the process of giving consideration to environmental protection and economic growth, we should pay attention to the coordination and complementarity between policy tools, and flexibly use environmental regulation tools in combination with market mechanisms. Gradually carry out policy reform and strengthen the applicability of environmental supervision in different fields.

**Author Contributions:** Conceptualization, Y.L., J.L. and Y.Z.; methodology, J.L.; software, J.L. and Y.Z; validation, J.L. and Y.Z; formal analysis, J.L.; investigation, Y.Z.; resources, Y.Z.; data curation, J.L.; writing—original draft preparation, J.L. and Y.Z.; writing—review and editing, Y.L., J.L. and Y.Z.; visualization, J.L.; supervision, Y.L. and Y.Z.; project administration, Y.L. and Y.Z.; funding acquisition, Y.L. and Y.Z. All authors have read and agreed to the published version of the manuscript.

**Funding:** This research was funded by Guangdong Provincial "13th Five-Year" Plan 2020 Discipline Co-construction Project, grant number GD20XTQ03 and Guangdong Higher Education Teaching Reform Project, grant number 2021SZ02.

**Institutional Review Board Statement:** Not applicable.

**Informed Consent Statement:** Not applicable.

**Data Availability Statement:** All data can be obtained by email from the corresponding author.

**Conflicts of Interest:** The authors declare no conflict of interest.

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
