# Peer review of "A Study on the Sustainable Relationship among the Green Finance, Environment Regulation and Green-Total-Factor Productivity in China"

_sustainability, doi:10.3390/su132111926_

Round 1
Reviewer 1 Report
Referee Report on “Green Finance and Regional Economic Quality: Construction of General Model with Environmental Regulation and Empirical Test”_ sustainability-1354163-peer-review-v2
This study used panel regression to discuss the relationship among the TFP (total-factor productivity), green finance and other macroeconomic variables. Direct effect and indirect effect (intermediary effect) models are used to explore the influence of the explained variables on the TFP. A relative performance of TFP are calculated by DEA-SBM, and panel regression of direct effect and indirect effect models are estimated. They found that the indirect effects of technology innovation and upgrading of industry structure exist significance influence on the TFP. Furthermore, the environment regulation has a nonlinear effect which impacts the TFP. The research methods in this article are basically appropriate and the results are useful. It is an interesting paper. However, I have the following specific concerns.
Major Concerns and Comments:
- The title is not fit of the article. I don’t understand why the TFP is used to represent the regional economic quality or High-Quality Economic development. The authors should define and explain. Construction and Empirical test in the title should be deleted. A new title, for example, ”A Study on the Relationship among the Green Finance, Environment Regulation and Total-factor productivity in China”.
- In table 4, column (1) and (2), FE and RE should be defined. From line 547 to line 565, column (2) is not mentioned or explained.
- In line 547, “Table 4. (1) shows the …” should be “In table 4, column (1) shows the …”. In line 566, “Table 4. (3)(4) shows…” should be “In table 4, columns (3) and (4) show…”. In line 567, “equation (3)” should be “columns (3)”, the same errors exist in line 570, 582, 583, 586, and 587-588.
- In conclusions, some description are not be discussed in the main body of this article, it should be deleted, for example “the development of green finance in China is generally in its infancy, and there is still great development potential. In space, it presents the gradient decreasing law of East→middle→West, coastal→inland, and there is a weak southwest northeast evolution trend. In the past 10 years, only Beijing has been in the first gradient, and the development of green finance is far ahead. Most areas are in the third and fourth gradient, and there is a large room for rise. Most of the areas in the lower gradient are northwest inland areas”.
- The authors should carefully check the format, symbols and spelling.
Evaluation:
I encourage authors to make submissions after making appropriate corrections.
Reviewer 2 Report
Dear authors,
Such kind of research is important to develop a carbon emission-free economy. However, the research has been containing many duplications and complicated issues that need to be simplified. For example, the introduction is huge and the data in the method is lacked in described quantitatively. Therefore, this research must be improved significantly before accept for publication.
- Abbreviation like EMB should be defined in the abstract
- What is carbon peak and carbon neutral? Needs to write in a clear way
- The method and the data use including the techniques must be elaborate in the abstract. A clear objective is also needed. The result has contained large sentences but the method i.e., the input data for the model has lacked in the abstract.
- Most of the time presenting roman numbers 1),2) …… is not habitual. Therefore, authors need to see re-writing the result again.
- The introduction is too large; sometimes there is duplication of issues, unnecessary issues such as COVID issues related to this research, line 177-178 and line 185-188 is duplication. The introduction must be coherent and clear for readers
- Figure 1 resolution and phone size is low
- How are the variables selected? and measurements? How can you become sure to be a “core explanatory variable”?
- The quantitative value for each variable must be provided by the authors because they should show how can construct a regression. Authors need to show the number of samples or points or pixels etc.
- The missing data caned obtained through the interpolation method. Can you show us each variable in spatial?
- Line 514- 516.” As shown in Figure 5, based on the development index of green finance in various regions, ArcGIS software is used to divide it into five gradients by using the natural discontinuity method to explore the development of green finance in China from the spatial level.”. It must be method; not result.
- If you use ArcGIS, the version must be added to the manuscript. Authors need to elaborate on the data types analyzed in the ArcGIS.
- Your data sources are many. Which data from which institution. Most of your data are from secondary sources why not test by yourself to validate the data and your research become strong.
- The map drown are not standard! Not altitude, latitude; sometimes direction is not visible.
Reviewer 3 Report
This paper systematically arranges the theory of green finance and economic growth, empirically tests the action mechanism of Green Finance on high-quality economic development by constructing models such as intermediary effect, regulation effect and threshold effect, which has strong practicability and logic. However, there are some problems to be further improved as well. My detailed comments are as follows:
- There is at least one case error in the manuscript. For example, in line 38 on page 1, "developing countries" should be "Developing Countries". Proper nouns should be capitalized. Please check the manuscript carefully.
- The introduction lacks some very relevant references, such as the definition and development of green finance. What are the differences and connections between green finance and traditional finance? What is the market mechanism and application prospect of green finance?
- In line 80 of the introduction, the author mentioned that green finance is the extension of China's economy in the financial field under the digital background. It should be explained in detail and put more accurately.
- The first letter of column 3 of Table 2 shall be capitalized. Line 330 on page 8 should have a period when naming the picture.
- I wonder if the author can add more references to the existing knowledge in the introduction to the relationship between green finance and high-quality economic development. In the introduction, the author should emphasize the rationality of using green finance for high-quality economic development.
- Another obvious problem with this paper is the lack of enough experimentation to demonstrate the validity and applicability of the proposed method. The author needs to do more experiments with more angles and show them in this paper.
- The relevant parameters of the threshold effect model and their interpretation in equation 14 are not detailed enough, please supplement.
- The author has made a detailed and sufficient model construction, which is innovative and logical. If possible, explain whether the advantages of the whole model and its method are universal.
- Figure 5 on page 15 is blurred. Please replace the image with a higher resolution.
- The suggestion part of the article is too thin. Please put forward more specific and detailed suggestions in combination with the model and theory to enrich the contribution of the article. Authors need to emphasize the novel insights obtained from their study.
- Please run an accurate spell checker and read carefully each sentence.
In all ,I would be very glad to re-review the paper in greater depth once it has been edited because the subject is really interesting. Hope everything goes well.
Round 2
Reviewer 2 Report
Accepted
Author Response
thank you for your work!